# The Effects of Ultrasonic and Gamma Irradiation on the Flavor of Potato Wines Investigated by Sensory Omics

**DOI:** 10.3390/foods12152821

**Published:** 2023-07-25

**Authors:** Chunlei Tan, Liang Tao, Jing Xie, Zhijin Yu, Yang Tian, Cunchao Zhao

**Affiliations:** 1College of Food Science and Technology, Yunnan Agricultural University, Kunming 650201, China; tcl98316@163.com (C.T.); taowuliang@163.com (L.T.); jingxie0624@163.com (J.X.); yzj070517@163.com (Z.Y.); 2Engineering Research Center of Development and Utilization of Food and Drug Homologous Resources, Ministry of Education, Yunnan Agricultural University, Kunming 650201, China; 3Key Laboratory of Precision Nutrition and Personalized Food Manufacturing, Ministry of Education, Yunnan Agricultural University, Kunming 650201, China; 4National Research and Development Professional Center for Moringa Processing Technology, Yunnan Agricultural University, Kunming 650201, China; 5Pu’er University, Pu’er 665000, China

**Keywords:** potato wine, artificial aging, volatile components, GC-MS-O, GC-IMS

## Abstract

Aroma is one of the most fascinating and least-known mysteries of Baijiu research. The volatile compounds (VOCs) of potato wine were evaluated by sensory omics techniques in order to comprehend their overall flavor characteristics and investigate the effects of ultrasonic treatment and gamma irradiation therapy on the aroma of the wine. The findings revealed that a total of 14 flavor compounds were identified by GC-MS. Isoamyl alcohol, ethyl octanoate, and 1,1-diethoxyethane were the key aroma components, according to GC-O analysis. A total of 50 volatile substances were identified by GC-IMS. After being subjected to irradiation and ultrasonic treatment, the alcohol level of the potato wine reduced while the esters content increased. By calculating the relative odor activity value, a total of 29 aroma components were classified as key aroma compounds (ROAV > 1). According to the results of the sensory evaluation—fruity, Fen-flavor, and sweet—and the acceptability of the irradiated and ultrasonicated potato wine were improved. Therefore, the use of ultrasonic and irradiation therapy in potato wine, as well as the overall aroma building of potato wine, can be supported theoretically by this study.

## 1. Introduction

Potato (*Solanum tuberosun* L.) is a member of the Solanaceae, and is the fourth most important crop after maize, rice, and wheat [1,2,3]. China produces an excessive amount of potatoes, so the creation of new products for processing them is quite important. Since ancient times of “Boiling wine and discussing heroes”, Baijiu has played a significant role in traditional Chinese culture. However, newly distilled Baijiu often has a pungent odor, sour and poor taste, and generally needs aging treatment to balance the overall flavor [4,5]. Owing to the long time and high cost of the traditional natural aging process, artificial aging technology has become a research hotspot [6]. Artificial aging technologies are mainly divided into physical aging (ultrasonic, gamma irradiation, magnetic field, etc.), chemical aging (oxidation method and catalytic method), and biological aging [7]. Chemical aging and biological aging are less frequently utilized than physical aging due to the wine body’s pollution and technical complexity, thus, on account of the mature technology and non-pollution to the wine body, ultrasonic (US) and gamma irradiation (GI) are widely used to accelerate the aging of Baijiu. Zheng et al. [8] treated greengage wine with low-frequency ultrasound technology, it was found that US was effective to accelerate the aging process, where the acid and ester contents increased, and the fusel oil and alcohol contents decreased. Emilio C et al. [9] treated Baijiu with ultrasound and found that higher amplitude and treatment time improved the protein stability of Baijiu, and the clarification effect was better than that of bentonite. Jia et al. [10] observed that GI could reduce the amount of time that Feng-flavored Baijiu took to age and that the best results were obtained when the radiation intensity was 5.9 KGy. According to Audrey [11], who employed gamma rays as an accelerated physical ripening technique to mature yellow rice wine, the right dose of GI could improve the defects and taste of the wine. Consequently, it is crucial to research the aging of potato wine.

Aroma is one of the important qualities that determine the quality of Baijiu and has an important influence on consumers’ choices [12]. In relevant research on Baijiu aroma, Jia et al. [10] studied Feng-flavored Baijiu by using UPLC-Orbitrap-MS/MS, found 29 compounds related to aging, and demonstrated that irradiation resulted in a significant number of VOCs in Baijiu. By GC×GC-TOFMS, Wang et al. [13] concluded that γ-terpinene, α-phellandrene, longicyclene, α-pinene, and limonene were the five terpenoids with the highest content in Qingke Baijiu. As a new detection method, headspace gas chromatography-ion migration spectrometry (HS-GC-IMS) has been widely used in the detection of VOCs in food due to its advantages of simple operation, high sensitivity, fast response, and causing little damage to samples [14,15,16]. With the help of GC-IMS and multivariate statistical analysis, Cai et al. [17] were able to quickly and accurately identify 29 VOCs in samples made by various manufacturers of sauce-flavored Baijiu; the first occurrences of acetone, propanal, acrolein, and pentanal were found. He et al. [18] characterized the Baijiu distillation process using HS-GC-IMS. There are limited reports on the impact of various aging techniques on Baijiu flavor based on flavor omics technology, despite extensive study on the VOCs of Baijiu. The research on artificial aging is crucial for Baiju, but it is also crucial to investigate how aging affects the flavor of Baiju using food flavor omics technology for both modern Baijiu brewing and traditional grain Baijiu brewing.

In order to examine the VOCs in potato wine, this study used GC-IMS along with sensory assessment, E-nose, and GC-MS-O for the first time. To identify the VOCs in potato wine under different treatment conditions, principal component analysis (PCA) and partial least squares discriminant analysis (PLS-DA) were employed. The VOCs that significantly contributed to the sample were then chosen using the relative odor activity value (ROAV) and gas chromatography-olfactory determination (GC-O) methods. This study offers a theoretical basis for exploring potato wine’s overall flavor composition and the use of irradiation and ultrasonic technologies in its manufacture.

## 2. Materials and Methods

### 2.1. Sample Preparation

Fresh potatoes were purchased from Dongchuan County (Yunnan, China) and stored at 4 °C. Chen’s Wine Industry Chuxiong, China) sponsored the fermented koji. Sun’s [19] method of making potato wine was used as a guide and modified. In brief, the fresh potatoes were washed and sliced, steamed for 30 min to remove the hard core, and added the raw material’s weight of water for beating. When the temperature dropped to 30 °C, the wine koji was inoculated at a dosage of 0.7 kg/100 kg. The fermentation temperature was controlled between 28 °C and 30 °C, and the fermentation process lasted 30 days. After fermentation, the fermentation liquid was distilled at 90 °C. About 100 mL of the head wine, containing many harmful substances, such as head wine and formaldehyde, was discarded, and a potato-distilled wine sample was obtained through two distillations.

According to previous methods [8,10], the potato wine was treated by ultrasonic wave and gamma irradiation. The US method was to put the newly produced potato wine into 300 mL airtight glass bottles, then put it in an ultrasonic processor for ultrasonic treatment at 450 W and 40° C for 5 h, then store at 4 °C for later use. The GI methods are as follows: the newly produced potato wine samples were placed in a 2 L brown sample bottle and transported to Yunnan Nuclear Application Technology Co., Ltd. (Kunming, China) for gamma radiation treatment. The radiation dose was 2–3 KGy, and the radiation source used was Co60.

For convenience labeling, newly produced potato wine was denoted as Y1, ultrasonic-treated potato wine was denoted as Y2, and gamma-irradiated potato wine was denoted as Y3.

### 2.2. Chemicals

3-methylthiopropana (95%, CAS: 3268-49-3), methanol (≥99%, CAS: 67-56-1), tert-amyl alcohol (≥99%, CAS:75-85-4), ethyl hexanoate (≥99%, CAS:6378-65-0), 3-methyl-1-butanol (≥99%, CAS:123-51-3), ethyl acetate (≥99%, CAS:141-78-6), ethyl lactate (≥99%, CAS:97-64-3), and cyclohexanone standard (>99.99%) were purchased from China Aladdin Co., Ltd. (Shanghai, China).

### 2.3. E-Nose Analysis

According to the method described by Wang et al. [20], the PEN3 E-Nose system (AIRSENSE, Schwerin, Germany), which has 10 different metal oxide sensors, was used to test the samples. Appendix A displays the performance of the sensors. In short, after natural reheating, absorb 1 mL of the potato wine sample and place it in a 100 mL beaker. It is then equally mixed, covered with double plastic wrap, and allowed to remain at room temperature for 30 min before being tested on the machine. The sampling time was 1 s per group, and the needle was directly put into the headspace bottle holding the sample for analysis. The injection flow rate was 400 mL/min, the sensor self-cleaning duration was 80 s, the sensor return to zero time was 5 s, and the sample preparation time was 5 s. The sample period for the analysis was 80 s.

### 2.4. Analysis of Volatile Compounds by GC-MS-O

With a few minor modifications from Wang et al. [21], the GC-MS analysis was performed using an Agilent 6890 series gas chromatograph and an Agilent 5973 N-series mass spectrometry detector (Agilent Technologies, Inc., Santa Clara, CA, USA). The VOCs were extracted using a manual solid phase microextraction injector and a 50/30 μm DVB/CAR/PDMS fiber extractor. The extractor was aged for 5 min at the gas chromatograph inlet at 230 °C before use. Then, 5 mL potato wine was added into a 15 mL headspace extraction bottle, and the extraction head was inserted into the extraction bottle for 30 min after equilibrium at 60 °C for 5 min. At the end of extraction, the extraction head was removed and inserted into the gas chromatographic injector and desorbed at 250 °C for 5 min. A DB-1MS capillary column of fused quartz (60 m × 0.25 mm × 0.25 μm) (Agilent Technologies, Inc., Santa Clara, CA, USA) was used for gas phase analysis. The initial conditions of gas chromatograph were 40 °C for 2 min, then 5 °C·min^−1^ to 85 °C for 2 min, then 2 °C·min^−1^ to 110 °C for 2 min, then 4 °C·min^−1^ to 160 °C for 1 min. Finally, the temperature was increased to 230 °C at 10 °C·min^−1^ for 5 min. The carrier gas (He) flow rate 1.0 mL·min^−1^; pressure 50.5 kPa; The injection method was no shunt injection. The odor was quantitatively determined by ion monitoring by gas chromatography. The mass-selective detector operated at 70 eV using an electron shock mode; the ion source temperature was 230 °C; the interface temperature was 230 °C; the quality scanning range was *m*/*z* 40~400. 

The qualitative results of VOCs were compared with the NIST11 spectrum library, and the matching degree was more than 79%. For the quantitative analysis of VOCs, the internal semi-quantitative standard approach was chosen. Using cyclohexanone (99.999% purity) as the internal standard and the concentration of cyclohexanone for the internal target was 2.50 mg/kg. The quantitative calculation method of VOCs was the concentration of cyclohexanone multiplied by the ratio of the volatile peak area to the internal standard peak area.

GC-O analysis was performed on a gas chromatography-mass spectrometry system with an olfactory detection port (ODP3, Gerstel, Mulheim, Germany). The evaluation team consisted of 5 professionals (3 women and 2 men) trained in odor recognition. The sample was described for aroma and evaluated for intensity (1 = recognizable odor; 2 = distinct odor; 3 = strong odor), three parallel experiments were conducted on the same sample, and the final aroma intensity (AI) was the arithmetic mean of the five members.

### 2.5. Analysis of Volatile Compounds by GC-IMS

FlavourSpec^®^ obtained from G.A.S. was used for analysis (Gesellschaft fur Analytische Sensorsysteme mbH, Dortmund, Germany). Referring to the method of He et al. [18], the sample of potato wine placed at 1 mL room temperature was weighed and placed in a 20 mL headspace bottle, incubated at 60 °C for 10 min before sampling, and the incubation speed was 500 r/min. The injection needle temperature was 85 °C. The injection volume was 500 μL. Cleaning took place for 30 s using high-purity N2 (purity > 99.99%). For gas chromatography, an MXT-5 column (30 m × 0.53 mm × 1 mm, RESTEK, Bellefonte, PA, USA) was employed. The flow rates were 2 mL/min (0~2 min), 2 mL/min~10 mL/min (2~10 min), and 10 mL/min~100 mL/min (10~20 min). The column temperature was 60 °C. The entire procedure took 40 min. For ion migration spectrometry, a 5 cm drift tube was utilized; the tube’s linear voltage was 400 V/cm; the drift gas flow rate was 150 mL/min; and N2 was used as the carrier gas/drift gas, with a flow rate of 0–2 mL/min. 45 °C was the predetermined temperature.

The retention time and migration time corresponding to the characteristic peaks of GC-IMS components can be used for VOCs, and the instrument can be standardized by the linear retention index (RI) of each volatile compound [18,22]. By comparing the standard RI and drift time (ms) in GC-IMS NIST library, the VOCs were qualitatively analyzed. The intensity of VOCs was obtained according to the peak volume, and the content of VOCs was proportional to the peak intensity.

### 2.6. Calculation of Relative Odor Activity Value (ROAV)

The VOCs’ ROAV_max_ that contributed the most to the flavor of the sample was defined as 100, and the ROAV of other VOCs was calculated as follows [23]:$$ROAVi\approx 100\times \frac{Ci}{Cmax}\times \frac{Tmax}{Ti}$$
where Ci and Cmax were the relative content/(%) of each VOC and the relative content (%) of the VOCs that contributes the most to the overall flavor of the sample, respectively. Ti and Tmax were the sensory thresholds/(mg/L) of each VOC and the sensory thresholds/(mg/L) of the VOCs that contributed the most to the overall flavor of the sample. The ROAV of the VOCs generally increases in proportion to the contribution to the sample’s overall flavor, and the threshold was discovered by looking through relevant literature.

### 2.7. Sensory Quantitative Description

Sensory analysis has slightly modified the methodology, as reported by Wu et al. [24]. Three samples of potato wine were evaluated by sixteen sensory description-trained evaluations (ten men and six women, aged 20 to 25). After discussion, fruity (ethyl acetate), alcohol (Ethanol), potato (3-methylthiopropanal), cursive (daqu, fermented koji), sweet (honey), and fragrance were determined as aroma characteristics of the potato wine. Color, taste, liking, purity, and style were determined as sensory indexes of potato wine. Detailed information on the aroma evaluation criteria is shown in Appendix A.

Finally, 10 excellent performers were selected according to the simulation experiment for the final sensory evaluation experiment. Three potato wine samples (15 mL) were poured into clear glasses at room temperature and randomly submitted to the team members for sensory evaluation. Aroma intensity ranged from 1 (very weak) to 15 (very strong). Three replicates were performed for each group of samples, and the average score of each sample was taken for further analysis.

### 2.8. Data Processing

SPSS Statistics 27 (Umetrics Corporation; Umea, Sweden) was used for ANOVA analysis and significance analysis (*p* < 0.05), and the data were expressed as mean ± s. PCA and PLS-DA were performed on three samples using MetaboAnalyst 5.0 to analyze sample differences and look for differentially labeled compounds. Unscrambler X established a partial least squares regression (PLSR) model to determine associations between the VOCs and the sensory properties. The Wayne chart and heat map were drawn using TBtools, the E-nose data was collected and processed by built-in Winmuster software, and the radar map and correlation heat map were drawn using Origin 2021 software (Originlab Corporation, Northampton, MA, USA).

## 3. Results and Discussion

### 3.1. Analysis of E-Nose

E-nose technology can quickly evaluate food quality by simulating the human sense of smell, and has been widely used in the differentiation of wine [25]. As shown in Figure 1A, W5S, W1S, W1W, W2S, W2W, and W1C were particularly responsive to the VOCs of potato wines. The response values of Y2 and Y3 samples in the three sensors of W1C, W1S, and W2S decreased, indicating that the content of benzene, alcohol ether aldehydes, ketones, methane, short-chain alkane, and other aromatic compounds decreased. The results showed that the odor differences of the three samples came from small molecules of nitrogen, oxygen compounds, aromatic components, organic, inorganic sulfur substances, alcohol ether aldehydes, and ketones. The reason for the result is that the newly produced wine often contains some VOCs, such as formaldehyde, which prevents the wine’s various flavoring agents from balancing one another and produces a bad scent and flavor [26]. After ultrasonic and irradiation treatment of potato wine, it may accelerate the oxidation or volatilization reaction of these substances, so that the composition structure of the VOCs in the wine body is more reasonable and reduces the pungent odor of the wine. Xu et al. [27] reported that the change trend of natural aging of liquor was similar to this result.

PCA analysis results of E-Nose are shown in Figure 1B. The contribution rates of PC1 and PC2 were 81.1% and 18.9%, respectively, with a total contribution rate of 100%; it can be concluded that the data accurately reflects the information in the samples as a whole. The three samples can be distinguished by E-nose, in which the overall information of Y2 and Y3 overlaps, indicating that their overall flavor was more similar. The distance between Y1, Y2, and Y3 indicated that Y1 had distinct flavor characteristics compared with the other two kinds. As can be shown, the E-nose can tell the three varieties of potato wine, but it cannot tell which VOCs are changing between them.

### 3.2. The VOCs in Three Potato Wines Identified by GC-MS-O

In order to study the differences of specific VOCs in the three samples, GC-MS combined with GC-O was used to analyze the VOCs in the samples. Five alcohols, four esters, one alkane, and four benzenes were properly identified by GC-MS (Appendix A). As shown in the Venn diagram in Figure 1C, a total of 14 VOCs were identified, among which, 10, 12, and 12 different kinds of VOCs were perceived by Y1, Y2, and Y3, respectively, and 9 components were common components. The VOCs of the three samples were different, to varying degrees. From the analysis of changes in the type of VOCs, the type of VOCs was different among the three samples, and the type changes were mainly reflected in ethyl propionate. The Y1 sample had the least types of VOCs, with 10 types. Compared with the other two samples, ethyl propionate, 4-isopropyltoluene, 1,2,4,5-tetramethylbenzene, and 1,2,3,4-tetramethylbenzene in the Y1 sample were not detected. 1,2-dimethyl-3-ethylbenzene was the specific component of the Y1 sample. After US treatment and GI treatment, the types of VOCs in the Y2 and Y3 samples increased to 12, among them, 4-isopropyltoluene and 1,2,4,5-tetratoluene were unique compounds of Y2 and Y3, respectively. According to the analysis of the content change of VOCs, the content of VOCs detected in Y3 was significantly higher than that in the other two samples. Thus, it is clear that the GI can have a significant impact on the VOCs in potato wine. Jia et al. [10] observed that appropriate irradiation might enhance the flavor components of liquor by gamma irradiating Feng-flavor liquor.

Three samples were identified by GC-O combined with the aroma intensity method. As shown in Table 1, three types of VOCs were found to be prominent through GC-O smelling analysis, which were isoamyl alcohol (unpleasant smell, 1.0), ethyl octanoate (fruity, 1.0) and 1, 1-diethoxy-ethane (wine flavor, 1.5), chromatograms and structural formulas of three key VOCs are shown in Figure 1D. Wang et al. [28] reported that isoamyl alcohol is the alcohol with the highest relative content in potato, and it is also the main alcohol in purple potato wine. After ultrasonic and gamma-ray irradiation, the relative percentage of isoamyl alcohol in potato wine decreased, which improved the overall flavor of the wine body. Ethyl caprylate is a common ester substance in sweet wine, showing a fruity and brandy flavor [27]. The content of ethyl caprylate in the Y2 and Y3 samples was significantly increased, which was consistent with the results obtained by Lin et al. [29], who used GC-MS to determine the VOCs of potato wine. 1,1-diethoxy-ethane, also known as ethyl acetal, can give liquor an aging flavor and is generally used as an indicator of the aging degree of liquor. Acetaldehyde reacts with alcohol to generate ethyl acetal during the natural aging process of liquor, which improves the flavor. The ethyl acetal content in Y3 increased significantly, indicating that GI could accelerate the aging speed of wine. Audrey [11] obtained similar conclusions by irradiating rice wine with gamma rays.

### 3.3. The VOCs in Three Potato Wines Identified by GC-IMS

The volatile components of potato wine processed using various techniques were examined using the Reporter plug-in program in the analysis program LAV of GC-IMS. The red vertical line at abscissa 1.0 in Figure 2A is the reactive ion peak (reactive ion peak, normalized) [14]. Each point on either side of the peak corresponds to a volatile molecule, and the color’s intensity reflects the level of content. Red denotes a high concentration, whereas white denotes a low one. The darker the hue, the higher the concentration. The entire spectrum can reflect all of the sample’s volatile flavor component data. In order to observe the difference of VOCs in three types of potato wine more directly, Y1 was taken as the reference object, and the signal peak of the Y1 sample was deducted from the other spectra to obtain the difference spectra between other samples and Y1 (Figure 2A). Most VOCs concentrations had increased in Y2 and Y3 compared to Y1 concentrations. The change in Y3 was more pronounced than the change in Y2. Combined with the fingerprint (Figure 2B), it can be seen that the content of methyl thiopropional, p-cymene, isoterpinene, α-terpinene, dimethyl thioether, ethyl formate, ethyl butyrate, acetone, 3-methyl-butanal, 2-methyl-propional, propional, Z-3-hexenol, 1-amyl alcohol, 1-butanol, and other VOCs in Y1 was higher. The content of γ-terpinene, butyl acetate, ethyl valerate, 2-butyl ketone, and other VOCs in Y2 was higher. The content of propionic acid, acetic acid, methyl benzoate, ethyl capric acid, ethyl lactate, ethyl caproate, isoamyl acetate, isobutyl propionate, ethyl 3-methyl butyrate, isobutyl acetate, propyl acetate, ethyl isobutyrate, methyl acetate, propyl propionate, isoamyl propionate, methanol, acrolein, and other VOCs in Y3 was higher. Compared with Y1, the content and composition of VOCs in Y2 and Y3 samples changed significantly. Xu et al. [27] found that the content of liquor ester increased significantly in a short time through ultrasonic aging of Baijiu. Jia et al. [10] found that the contents of ethyl caprylate and ethyl caproate were significantly increased by γ-irradiation of newly brewed Fen-flavor liquor, which was consistent with the results of this study.

### 3.4. Differences of VOCs in Three Potato Wines by GC-IMS

Qualitative analysis of VOCs in three types of potato wine was conducted through the built-in NIST and IMS databases of GC-IMS. As shown in Figure 3B, a total of 53 types of volatile compounds were detected in three potato wines, among which 50 types were accurately characterized. Appendix A illustrates the retention index, migration time, and peak-volume information for all drugs in each sample. Among them, 3-methyl-1-butanol, 1-butanol, 2-methyl-1-propanol, and 3-methylbutyl propanoate exist both as monomers and dimers. 

Appendix A shows that the three samples’ peak area intensities were in the following order: Y3 > Y2 > Y1, which indicated that the content of VOCs in potato wine was improved after ultrasonic and irradiation treatment, and the effect of GI was greater than that of US, which is consistent with the results of Jia et al. [10], who found a significant increase in the VOCs in the body of Fen-flavored liquor by γ-ray irradiation. The types of VOCs in the three samples remained constant, but their contents significantly altered (as shown in Appendix A and Figure 3A). A total of 2 acids, 8 aldehydes and ketones, 14 alcohols, 21 esters, and 5 other substances were detected. As shown in Figure 3A, compared with Y1, the relative contents of alcohols, aldehydes, and ketones in Y2 and Y3 decreased, while the relative contents of esters in Y2 and Y3 increased significantly.

Alcohol was the highest proportion of VOCs in potato wines, with a relative content of each sample being 61.05% to 64.10%, and the highest content of ethanol was 24.85% to 25.82%. Relevant studies show that ethanol and water are the main components of Baijiu, accounting for about 98% of the total amount [19]. In addition to ethanol, 2-methyl-1-propyl alcohol and 3-methyl-1-butanol had higher content, and there were monomers and dimers, respectively; these two alcohols are the senior alcohols. In addition to ruining the flavor of wine, too much of these two alcohols can be harmful to human health [30]. After ultrasonic and irradiation treatment, the relative content of the two alcohols decreased, which may be caused by the two treatments accelerating the oxidation reaction of the alcohols. 1-Amyl alcohol, which is considered to be a characteristic component in potato wine, was also detected in this experiment.

Ester compounds were the most varied VOCs in potato wine, and their content was second only to alcohol in potato wine, ranging from 28.38% to 31.44% in each sample. After treatment, Y2 and Y3 had a higher ester content than Y1, which may be caused by chemical esterification of fatty acids interacting with the ethanol. Ethyl acetate, isoamyl acetate, ethyl propanoate, and ethyl butanoate ranked in the top four. Ethyl Acetate is one of the key VOCs of Fen-flavored Baijiu, so potato wine should be regarded as a Fen-flavored Baijiu. Among them, 11 kinds of esters were mostly ethyl esters, and 4 types of methyl esters, 3 types of butyl esters, 2 types of propyl esters, and 1 type of amyl ester were detected. Esters, which are primarily created through the esterification process between alcohols and carboxylic acids, are thought to be the most significant trace elements in determining the scent of liquor. In addition, yeast and some microorganisms can also produce esters during the fermentation process [31].

Aldehydes and ketones present nutty and fruity aromas, which can enhance the aroma and taste of liquor in the proper concentration range [27]. The contents of aldehydes and ketones in each wine sample ranged from 5.50% to 6.67%, and most of the volatile aldehydes and ketones were metabolized by yeast cells during fermentation [12]. Moreover, after ultrasonic and irradiation treatment, the content of aldehydes and ketones in Y2 and Y3 decreased, which may be the reason why aldehydes and ketones are easy to oxidize, or because the increase accelerated the volatilization rate of both. 

Only acetic acid and propionic acid were detected. Acetic acid and propionic acid are widely found in various types of wine. Acid compounds can maintain ester aroma, balance wine flavor, and coordinate aroma [32]. Acetic acid is mainly produced by the ethanol metabolic pathway. There is a particular equilibrium between the acids and esters in different types of liquor, according to numerous studies on the flavor of liquor. The concentration of the acid determines the concentration of the ester. As a result, the balance of acid and ester will become a crucial control point for the quality of liquor [12,18].

This investigation also identified five other kinds of chemicals, including terpinolene, gamma-terpinene, alpha-terpinene, p-cymene, and dimethyl sulfide. Among them, p-cymene (also known as 4-isopropyl toluene) was also detected by GC-MS. Gamma-terpene and alpha-terpinene are the signature aroma substances in prickly ash, which were first reported in potato wine.

### 3.5. Multivariate Statistical Analysis by GC-IMS

To further understand the differences between the three potato wines, the PCA model and PLS-DA model were established to distinguish the samples. PCA can explain the differences between multi-dimensional samples and further obtain the critical comprehensive analysis of samples with a large number of complex relational factors [22]. According to Figure 4A, the contribution rate of PC1 and PC2 was 93.6% and 4.6%, respectively, and the cumulative contribution rate was 98.2%. Y1 had a negative response value to PC1, and Y2 and Y3 had a positive response value to PC1. There were clear differences between the three samples. The samples of Y2 and Y3 were located on the positive semi-axis of PC1, and their proximity to one another suggests that the two samples shared some characteristics. After ultrasonic and irradiation treatment, similar effects will be produced on the flavor of potato wine. At the same time, Y2 was located in the negative semi-axis of PC2, while Y3 was located in the positive semi-axis of PC2, which proved that there was a certain gap between Y2 and Y3 samples. The loading plot in Figure 4C allows a clearer study of the differences between the compounds in the three samples, demonstrating the association between the volatile compounds and the samples [20]. V27 (ethanol), V10 (3-methyl-1-butanol-D), and V17 (2-methyl-1-propanol-D) are located in PC1, and their responses were negative and may be classified as Y1. Y2 was in the fourth quadrant with positive PC1 response values, and Y2 was primarily positively connected with V14 (isoamyl acetate) and V22 (isobutyl acetate); ultrasound treatment had a larger effect on both V14 (isoamyl acetate) and V22 (isobutyl acetate) in potato wine.

PLS-DA model was used to distinguish the three wines, and the flavor compounds with significant differences (VIP > 1) among the three wines were looked for. R^2^ was higher than 0.97 and Q^2^ was higher than 0.93, indicating that the model was excellent. In Figure 4E, after 1000 replacement tests, the model *p* value was 0.066, indicating the high reliability of the model. Figure 4B shows the classification pattern of potato wine using the PLS-DA model, with variances explaining 93.6% and 4.5% of components 1 and 2, respectively. Y1 was significantly separated from Y2 and Y3 in component 1, which was consistent with PCA and E-nose analysis results. PLS-DA could be used to distinguish the three potato wines well. Finally, variable importance projection (VIP) plots were used to screen VOCs with significant differences. Overall, compounds containing VIP > 1 are considered to be VOCs with significant differences [33]. As shown in Figure 4D, isoamyl acetate, 3-methyl-1-butanol-D, 2-methyl-1-propanol-D, isobutyl acetate, ethanol, ethyl acetate, and propanal were the seven compounds with the highest degree of difference (VIP > 1). Then, a cluster analysis was conducted on the wine samples using differential VOCs, as shown in Appendix A. The colors from blue to red represent the relative content from low to high [22]. The clustering of the wine samples using differential compounds found that the samples were divided into two groups, with Y2 and Y3 grouped into one group and Y1 in its own group (Appendix A). The wine body’s composition changed overall following US and GI treatment, and the various compounds were classified into four clusters (Appendix A), among which the relative contents of the esters in Y2 and Y3 were higher, such as isoamyl acetate and isobutyl acetate, etc. 3-methyl-1-butanol-D, 2-methyl-1-propanol-D, ethanol, and propanal were higher in Y1. According to the study by Xu et al. [31], the formation of esters depends on the supply of acyl CoA and alcohols generated in the process of β-oxidation, and alcohol acyl transferase (AA-Ts) can combine various alcohols and acyl CoA to synthesize a variety of esters. Us and GI may accelerate the process of alcohol conversion to esters and thus affect the conversion of ethyl acetate to acetic acid.

### 3.6. The Key-VOCs Analysis by ROAV

The concentration of compounds is not directly related to the flavor characteristics, which are mainly affected by the sensory threshold of the VOCs [25]; therefore, the key aroma components in three samples were studied. Since the threshold was low and the relative concentration of isoamyl acetate in the samples was high, the ROAV for isoamyl acetate was adjusted to ROAV_stan_ (=100). In the three samples, 29 VOCs were found to be key aroma compounds (ROAV > 1) (Appendix A), including 14 esters, 8 alcohols, and 5 aldehydes. Among them, propanal (35.70~68.30), 2-methyl-propanal (20.30~31.38), 3-methyl-1-butanol-D (112.00~156.51), ethyl octanoate (138.45~175.00), isoamyl ethyl butanoate (100), ethyl butanoate (204.75~301.13), isobutyl acetate (46.77~49.93), ethyl compounds such as isobutyrate (105.58~111.73), and p-cymene (28~101.53) had high ROAV values. These VOCs contribute significantly to the aroma of potato wines. Among these components, 2-methyl-propanal, ethyl octanoate, ethyl isobutyrate, and p-cymene had low concentrations but relatively high ROAV due to the very low threshold of this component. It can be seen from Appendix A, compounds with ROAV values greater than 1 in the three samples did not change, but the ROAV values of VOCs did. The ROAV values of aldehydes, ketones, and alcohols decreased in potato wines, while the ROAV values of ethyl octanoate, isobutyl propionate, and ethyl hexanoate were increased, which was related to the change in the relative content of VOCs in wines. Esters are thought to be the primary factors that affect the aroma characteristics of the wine body because they often have a high ROAV value and a low aroma threshold. Esters mainly provide fruity, floral, sweet, and milky flavors in Baijiu. For example, ethyl acetate (ROAV; 1.97~2.67), with a fresh fruit aroma, and ethyl butyrate (ROAV; 2.91~3.30) have an odor similar to pineapple and banana [31], both of which will give the Baijiu a clean and fresh flavor. Ethyl valerate (ROAV; 4.91~7.86), ethyl propionate (ROAV; 1.55~1.98), ethyl lactate (ROAV; 0.03) has a fruity and slightly fatty flavor, which is often detected in Maotai-flavored Baijiu [34]. Ethyl caproate (ROAV; 7.03~14.00), with a fruit aroma, is the most important flavor ester in Luzhou-flavored Baijiu, isoamyl acetate (ROAV; 100), which has a pear and apple aroma and is present in large concentrations, causes wine to have a terrible smell [31]. Higher alcohols are the products of amino acid or sugar metabolism in the fermentation process of yeast alcohol, which mainly produces fragrant wine, fruit, and floral aromas. At the same time, they are also the precursor substances of esters, which play an important role in the flavor of Baijiu [35]. However, some higher alcohols, such as 3-methyl-1-butanol (ROAV; 19.12~156.51), have a nail-polish flavor, which leads to the overweight taste of highland barley wine and beer [36]. In addition, it has been demonstrated that 2-methyl-1-propyl alcohol (ROAV; 3.04~10.63) and 2-butanol (ROAV; 8.78~11.26) cause bitter tastes and so impact taste [37]. 3-methylbutanal (ROAV; 1.05~1.85) is considered a potential source of grass fragrance in Chinese Jing Jiu [12]. Alpha-terpene (ROAV; 1.77~4.62) and P-cymene (ROAV; 28~101.53) have a lemon aroma and woody aroma, respectively, which have an important influence on wine sensory [21], and were also detected in potato wine this time.

### 3.7. Sensory Quantitative Description of Different Potato Wines

Quantitative sensory description and analysis were carried out on three potato wines. The evaluation results are shown in Table 2. The significance analysis showed that there were significant differences in fruity, sweet, style, and preference among the three potato wine samples (*p* ≤ 0.001), followed by alcohols, whose taste attributes were extremely significant (*p* ≤ 0.01); cooked potato flavor and sweet flavor attributes were also significantly different (*p* ≤ 0.05). As shown in Table 2, the average scores of six aroma attributes of the three potato wines with different treatment methods were significantly different, and the three aroma attributes of alcohols, Fen-flavor, and fruity have a greater impact on aroma characteristics of potato wines, with a higher score. After US and GI treatment, the properties of fruity, sweet, and Fen-flavor of potato wine were improved, while the scores of alcohols, sweet and cooked potato were decreased correspondingly, which was consistent with the results obtained by ROAV value. In terms of sensory properties, the tasted, style and preference of the treated potato wines were higher than those of the newly produced potato wine, and the color and purity scores were similar. In general, the scores of sensory attributes of US and GI were more similar, especially the sensory attributes of style and preference. However, there were also differences between the two samples. For example, the score of the Y2 sample on tasted (13.20) was higher than that of the Y3 (12.85) and Y1 (12.15) methods. The Y3 sample scored high in the sensory attributes of style (13.03) and preference (13.25), indicating that the taste, style, and preference of the potato wine body could be improved by US or GI treatment.

### 3.8. Correlation Analysis of VOCs with Sensory Properties and E-Nose

The partial least squares regression (PLSR) model was established to better understand the relationship between 29 VOCs (ROAV ≥ 1) and 8 aroma sensory properties of potato wines. As shown in Figure 5, the two principal component explanatory variables of the PLSR model were 89% of the x-variable (VOCs) and 99% of the y-variable (aroma sensory attributes), and all aroma attributes and VOCs were located between the two ovals, indicating that the model had good explanatory ability [38]. The main correlation substances of cooked potato and alcohol aroma were 2-methyl-1-propanol-D, 1-butanol, alpha-terpinene, and ethyl butanoate. Combined with Table 2 and Appendix A, the content of these four substances in Y2 and Y3 was lower than that in Y1. At the same time, Y1 had higher cooked potato and alcohol properties than Y2 and Y3. Fruity, Fen-flavor, sweet, preference, and style were positively correlated with 1-hexanol, ethyl octanoate, ethyl hexanoate, isoamyl acetate, isobutyl acetate, and ethyl pentanoate; most of these VOCs were esters. 2-butanone and propyl acetate were associated with taste. Combined with the heat map of sensory properties and VOCs of potato wine, as shown in Figure 6A, the sensory properties of fruity and alcohols showed a significant positive correlation with p-cymene, 3-methyl-1-butanol-D, alpha-terpinene, 1-butanol-M,2-methyl-1-propanol-D, 2-methyl-1-propanol-M, and ethyl butanoate, as most alcohols and esters in liquor were characterized by a fruity and mellow flavor [14]. Ethyl octanoate, ethyl hexanoate, isoamyl acetate, isobutyl propionate, ethyl 3-methylbutanoate, isobutyl propionate, ethyl octanoate, ethyl hexanoate, isoamyl acetate, isobutyl acetate, propel acetate, 2-butanol, ethyl isobutyrate, and ethyl propanoate were positively correlated with sweet, Fen-flavor, tasted, style, and preference; this showed that esters cannot only provide sweet smell in the wine body, but also affect the taste of the wine body, and further play a key role in the wine style and people’s preference.

Figure 6B showed the link between the electronic nose and key flavor compounds, we can see that p-cymene, 3-methyl-1-butanol-M, alpha-terpinene, 1-butanol-M, 2-methyl-1-propanol-M, and 2-methyl-1-propanol-D showed a significant positive correlation with W2S, W1C, and W1S; isobutyl propionate, ethyl 3-methyl butanoate, isobutyl acetate, 2-butanol, propel acetate, isobutyl propionate, ethyl 3-methyl butanoate, 2-butanol, propel acetate, ethyl propanoate, and ethyl isobutyrate were significantly positively correlated with W1S, W1W, and W2W. This indicated that the electronic nasal receptors have different sensitivity to different substances. W1S, W2S, and W1C are more sensitive to alcohols and ketones, while W1S, W1W, and W2W are more sensitive to esters. The correlation analysis further indicated that ultrasonic treatment and irradiation treatment could improve the taste and aroma of potato wine by changing the content of volatile substances. In conclusion, the combination of sensory evaluation, the electronic nose and key aroma analysis can reveal the flavor information of potato wine with different treatments in a more comprehensive and intuitive way, help distinguish potato wine with different treatments, and intuitively explain the causes of aroma differences [39].

## 4. Conclusions

In this study, the aroma composition of potato wine was investigated using E-nose, GC-MS-O, and GC-IMS, and the effects of US and GI treatment on potato wine were discovered. The results showed that both US and GI treatment could influence the VOCs of potato wine. E-nose can identify and distinguish three kinds of potato wine well. Fourteen VOCs were identified by GC-MS in the three samples. Ten VOCs were present in Y1, twelve in Y2, and twelve in Y3, respectively. The effects of GI were superior to US, and the concentrations of VOCs in potato wine increased following ultrasonic therapy and irradiation treatment. According to the GC-O strength test, isoamyl alcohol, ethyl caprylate, and 1, 1-diethoxy-ethane all played a significant role in the VOCs of potato wine. A total of 50 VOCs were identified by GC-IMS, including 2 acids, 8 aldehydes, 14 alcohols, 21 esters, and 5 other compounds. Consistent with the results of GC-MS, the GC-IMS detection showed that US and GI could increase the aroma substance content of potato wine. The calculation showed that compared with Y1, the relative contents of alcohols in Y2 and Y3 samples decreased while the relative contents of some esters increased. Three kinds of potato wine were distinguished by the PCA and PLS-DA models, and seven different VOCs were selected as isoamyl acetate, 3-methyl-1-butanol-D, and 2-methyl-1-propanol-D (VIP > 1). Twenty-nine key VOCs, including ethyl butanoate and isoamyl acetate (ROAV > 1) were determined by ROAV value, and fourteen of them were esters. The sensory results showed that the aroma of potato wine was mainly Fen-flavor, fruity, and alcohols. After US and GI technology, the alcohols, daqu aroma, and cooked potato of potato wine decreased, while the Fen-flavor, fruity, and sweet of the potato wine were improved. Furthermore, the taste, style, and preference of the potato wine have been improved. In general, this study provides an effective reference for the construction of potato wine aroma maps and the application of ultrasonic technology and gamma irradiation technology in improving potato wine aroma.

## Figures and Tables

**Figure 1 foods-12-02821-f001:**
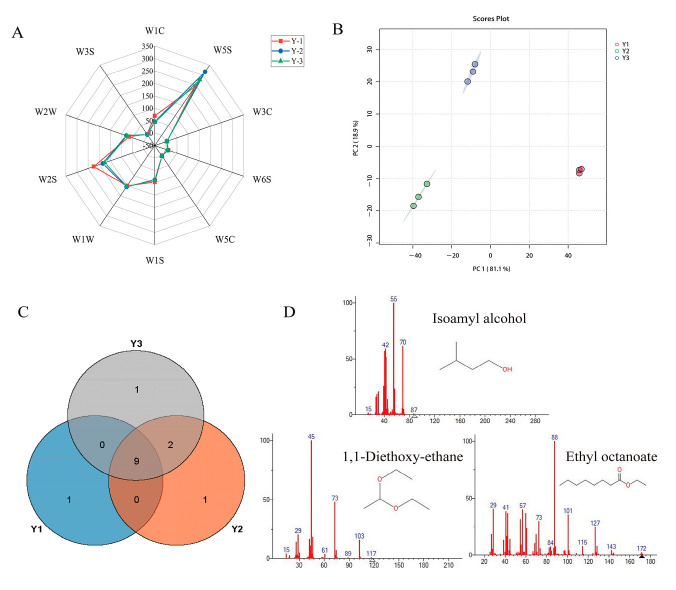
Radar chart (**A**) and principal component analysis (**B**) of the electronic nose data for three potato wines. The Venn diagram of 14 volatile compounds in different types of potato wine by GC-MS-O (**C**), ion diagrams and structural formulas of four volatile compounds by GC-MS-O (**D**). Y1: newly produced potato wine; Y2: ultrasonic treated potato wine; Y3: gamma-irradiated potato wine.

**Figure 2 foods-12-02821-f002:**
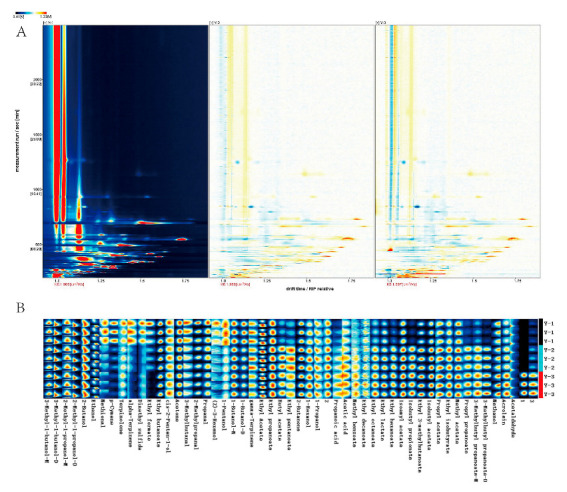
(**A**) Differentiation plot of volatile compounds. In Y2 and Y3, red and blue dots indicates that the concentration of the compounds are higher and lower than the reference (Y1), respectively, (**B**) gallery plot fingerprint of different potato wines by GC-IMS. Y1: newly produced potato wine; Y2: ultrasonic treated potato wine; Y3: gamma irradiated potato wine.

**Figure 3 foods-12-02821-f003:**
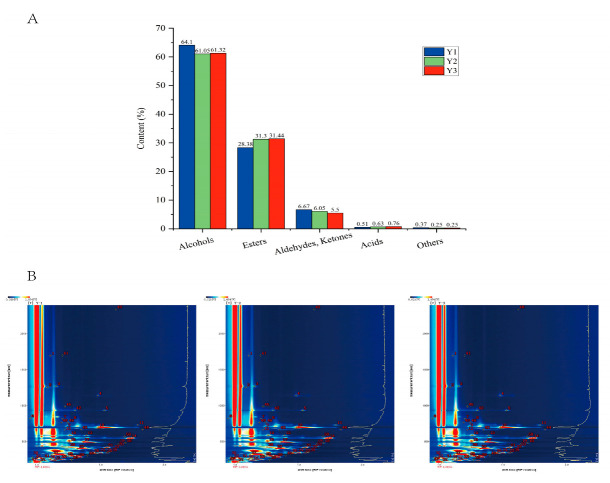
(**A**) The content of volatiles of different categories identified by GC–IMS in three potato wine samples. (**B**) Topographic plots of GC–IMS spectra with the selected markers obtained with different parts of Y1, Y2, and Y3. Y1: newly produced potato wine; Y2: ultrasonic treated potato wine; Y3: gamma irradiated potato wine.

**Figure 4 foods-12-02821-f004:**
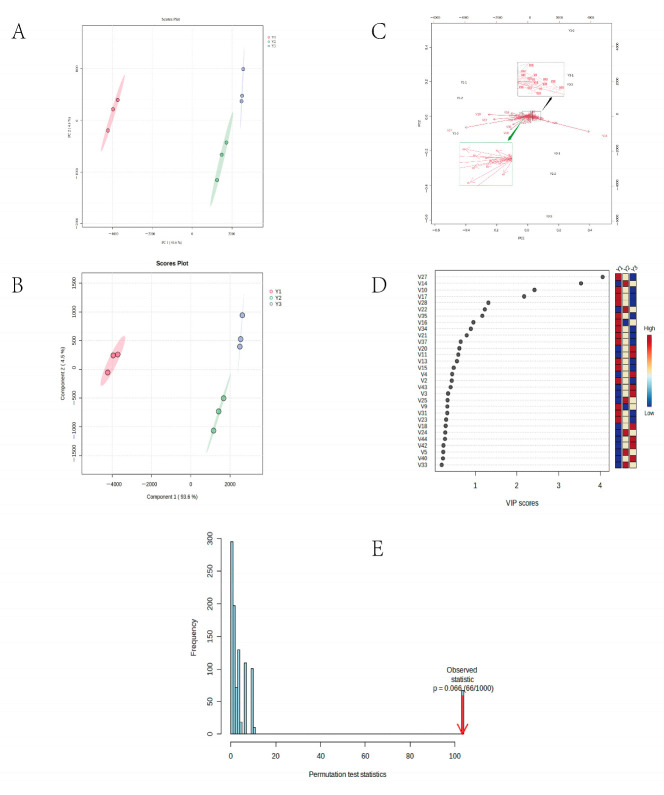
Multivariate analysis of aroma compounds in three potato wines based on GC–IMS; (**A**,**B**) are the 2D scores by PCA and PLS-DA, respectively; (**C**) is the loading plots analyzed by PCA, (**D**) is a graph of the VIP score for volatile compounds; (**E**) is a permutation plot tested 1000 times. The number is consistent with Appendix A, Y1: newly produced potato wine; Y2: ultrasonic treated potato wine; Y3: gamma irradiated potato wine.

**Figure 5 foods-12-02821-f005:**
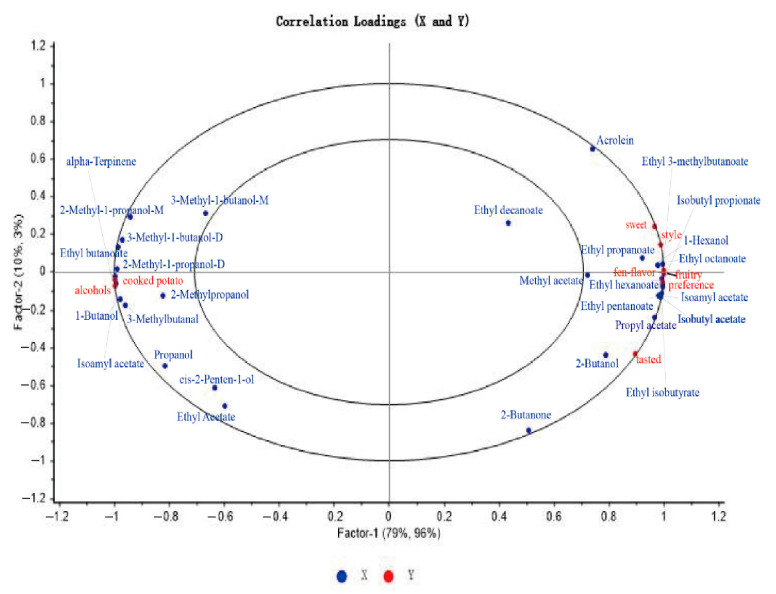
Correlation loading plot for aroma-active compounds (X-matrix) and aroma attributes of potato wines (Y-matrix). Blue and red represent the VOCs and aroma sensory properties, respectively.

**Figure 6 foods-12-02821-f006:**
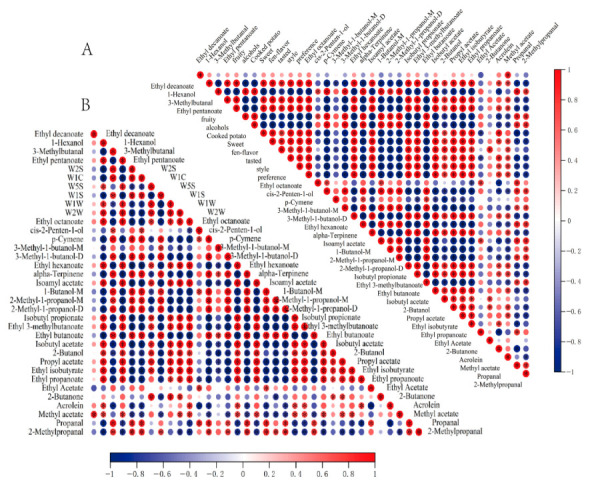
Pearson’s correlation analysis between the aroma-active compounds and aroma attributes (**A**), as well as the correlation between aroma-active compounds and E-nose (**B**). The large circle indicated a strong correlation, whereas the small circle indicated a weak correlation. The color bar denoted the R-value of the Pearson correlation, with 1 representing a perfect positive correlation (dark red) and −1 representing a perfect negative correlation (dark blue). The “*” in the circle indicated the aroma-active compounds were significantly correlated with the aroma attributes and E-nose (*p* < 0.05).

**Table 1 foods-12-02821-t001:** Results of GC-O of three kinds of potato wine.

Compound	GAS	Concentration (mg/kg)			Aroma Description	AI
		Y1	Y2	Y3		
Ethyl alcohol	64-17-5	24,760.76 ± 68.90	18,042.32 ± 71.23	44,814.64 ± 16.40	ND	ND
N-propanol	71-23-8	105.32 ± 4.64	75.12 ± 4.96	203.05 ± 6.25	ND	ND
Isobutyl alcohol	78-83-1	308.20 ± 6.23	189.18 ± 9.78	353.88 ± 21.53	ND	ND
(R)-(-)-2-butanol	14898-79-4	190.32 ± 10.09	157.15 ± 6.91	335.72 ± 5.30	ND	ND
Isoamyl alcohol	123-51-3	1152.53 ± 9.81	678.77 ± 4.66	1276.58 ± 24.72	unpleasant smell	1.0
Ethyl acetate	141-78-6	476.29 ± 25.04	381.93 ± 8.47	584.15 ± 10.57	ND	ND
Ethyl octanoate	106-32-1	96.40 ± 3.99	136.61 ± 3.82	154.83 ± 4.44	fruity	1.0
Ethyl decanoate	110-38-3	482.71 ± 9.65	326.2 ± 5.22	456.38 ± 5.92	ND	ND
Ethyl propionate	105-37-3	ND	44.76 ± 4.06	68.64 ± 1.61	ND	ND
1,1-diethoxy-ethane	105-57-7	90.87 ± 1.20	87.56 ± 0.52	161.58 ± 0.73	wine flavor	1.5
1,2-dimethyl-3-ethylbenzene	933-98-2	20.55 ± 0.44	ND	ND	ND	ND
4-isopropyltoluene	99-87-6	ND	22.68 ± 0.56	ND	ND	ND
1,2,4,5-tetramethylbenzene	95-93-2	ND	ND	18.98 ± 0.99	ND	ND
1,2,3,4-tetramethylbenzene	488-23-3	ND	21.94 ± 0.11	21.62 ± 0.51	ND	ND

Results are expressed as average (*n* = 3) ± standard deviation. Y1: newly produced potato wine; Y2: ultrasonic treated potato wine; Y3: gamma irradiated potato wine; ND: not detected in sample; AI: representative aroma intensity.

**Table 2 foods-12-02821-t002:** Average Score of aroma intensity under Three Potato wines.

Aroma Attribute	Mean Score of the Sample			Significance (*p* Value)
	Y1	Y2	Y3	
fruity ***	6.85 ± 1.07 ^b^	9.24 ± 1.43 ^a^	9.67 ± 1.30 ^a^	≤0.001
alcohols **	12.41 ± 1.07 ^a^	10.75 ± 1.23 ^b^	10.30 ± 1.41 ^b^	≤0.01
cooked potato *	5.72 ± 1.39 ^a^	4.45 ± 1.10 ^b^	4.17 ± 0.93 ^b^	≤0.05
sweet *	6.85 ± 1.43 ^b^	8.18 ± 1.59 ^ab^	8.99 ± 1.70 ^a^	≤0.05
Fen-flavor ***	7.48 ± 0.94 ^b^	9.33 ± 1.10 ^a^	9.61 ± 0.85 ^a^	≤0.001
daqu aroma	6.95 ± 0.88 ^a^	5.65 ± 1.73 ^b^	6.15 ± 1.23 ^ab^	>0.05
color	13.13 ± 0.49 ^a^	13.35 ± 0.58 ^a^	13.35 ± 0.47 ^a^	>0.05
tasted **	12.15 ± 0.75 ^b^	13.20 ± 0.48 ^a^	12.85 ± 0.58 ^a^	≤0.01
style ***	11.98 ± 0.56 ^b^	12.73 ± 0.37 ^a^	13.03 ± 0.59 ^a^	≤0.001
Preference ***	11.96 ± 0.78 ^b^	13.14 ± 0.32 ^a^	13.25 ± 0.49 ^a^	≤0.001
purity	13.35 ± 0.58 ^a^	13.12 ± 0.27 ^a^	13.09 ± 0.37 ^a^	>0.05

Results are expressed as average (n = 3) ± standard deviation. ^a–b^ Mean ± standard deviation with different lowercase letters on the same line differed by the Tukey test (*p* < 0.05) between treatments. Levels of significance: *, significant (*p* ≤ 0.05); **, highly significant (*p* ≤ 0.01); and ***, very highly significant (*p* ≤ 0.001). Y1: newly produced potato wine; Y2: ultrasonic treated potato wine; Y3: gamma irradiated potato wine.

## Data Availability

The data used to support the findings of this study can be made available by the corresponding author upon request.

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
