# Peer review of "The Effects of Ultrasonic and Gamma Irradiation on the Flavor of Potato Wines Investigated by Sensory Omics"

_foods, 2023, doi:10.3390/foods12152821_

Round 1

Reviewer 1 Report

The manuscript address an interesting topic which could bring new insights about the role of ultrasonics and gamma irradiation to the sensory profile of potato wines. 

However, in its current form, it feels difficult to follow. It is very easy to lose track. There is an undoubtedly amount of work, but it feels that each section almost immediately jumps to the next, with a new figure, of a new method, without getting into the details and highlighting the outcomes of the previous ones.  A schematic flowchart where authors specify all type of analyses, methods and type of plots, will help the reader. 

Figures should always be shelf explanatory, but in this case, there is a lack of having a bit more explanation in the text. 

Specific comments

Is it correct that SPSS is from Umetrics?

In the results section, few words appear with a - in between. Please check.

Line 225 "Irritation of the wine" does not sound a correct way of referring. check. 

Explain Fig1C a bit more.

Line 244. "The number of VOCs in Y1 244 sample was the least, which was 11 types." Rephrase

Even if shelf explanatory, it would be better to explain Figure 2 a bit more. 

For the PLS-DA, can you specify what are the 2 different matrices of data, VOCs obtained from GC-IMS, and?

It is explained in Section 3.8, but not clear for the PLS-DA of the previous section.

The English could be improved

Reviewer 2 Report

very interesting approach to the artificial maturation, studied throught chemical composition change and sensorial measurments.

But I think that without comparing artificial aged wine to natural ones, it cannot be improved.

Please check the PDF for further comments.

Round 2

Reviewer 1 Report

The manuscript has improved with the additions and modifications.